# Clinical Profile of Patients with Primary Sjögren’s Syndrome with Non-Identified Antinuclear Autoantibodies

**DOI:** 10.3390/diagnostics14090935

**Published:** 2024-04-30

**Authors:** Dorian Parisis, Julie Sarrand, Xavier Cabrol, Christine Delporte, Muhammad S. Soyfoo

**Affiliations:** 1Department of Rheumatology, Erasme Hospital, Hôpital Universitaire de Bruxelles (HUB), Université Libre de Bruxelles, 1070 Brussels, Belgium; dorian.parisis@ulb.be (D.P.); julie.sarrand@ulb.be (J.S.); xavier.cabrol@ulb.be (X.C.); 2Laboratory of Pathophysiological and Nutritional Biochemistry, Faculty of Medicine, Université Libre de Bruxelles, 1070 Brussels, Belgium; christine.delporte@ulb.be

**Keywords:** Sjögren’s syndrome, autoantibodies, rheumatoid factor, systemic manifestations, corticosteroids

## Abstract

Objectives—The aim of the present study was to characterize the clinical phenotype of patients with primary Sjögren’s syndrome (pSS) with non-identified antinuclear antibodies (ANA) in comparison with that of patients with pSS with negative ANA, positive typical ANA (anti-Ro/SSA and/or La/SSB) and positive atypical ANA. Methods—We conducted an observational, retrospective monocentric study at the Erasme University Hospital (Brussels, Belgium). Two hundred and thirty-three patients fulfilling the 2002 American–European Consensus Group criteria for pSS were included in this study. The patients were subdivided according to their ANA profile and demographics. The clinical and biological data of each subgroup were compared. Moreover, the relationships between these data and the ANA profiles were determined by multiple correspondence analysis. Results—In our cohort, 42 patients (18%) presented a non-identified ANA-positive profile. No statistically significant difference could be observed between non-identified ANA patients and ANA-negative patients in terms of age and/or ESSDAI score at diagnosis. There were significantly more frequent articular manifestations, positive rheumatoid factor (RF), and the use of corticosteroids in anti-Ro/SSA-positive patients compared to ANA-negative (*p* ≤ 0.0001) and non-identified ANA-positive patients (*p* ≤ 0.01). However, a significantly higher proportion of RF positivity and corticosteroid treatment was observed in non-identified ANA-positive patients compared to ANA-negative patients (*p* < 0.05). Conclusions—For the first time to our knowledge, our study has characterized the clinical phenotype of patients with pSS with non-identified ANA at diagnosis. The non-identified ANA-positive patients featured mostly a clinical phenotype similar to that of the ANA-negative patients. On the other hand, the non-identified ANA-positive patients were mainly distinguished from the ANA-negative patients by a greater proportion of RF positivity and the need for corticosteroid use due to articular involvement.

## 1. Introduction

Primary Sjögren’s syndrome (pSS) is a chronic systemic autoimmune disease characterized by lymphoplasmacytic infiltration of the exocrine glands—mainly salivary and lacrimal—responsible for sicca syndrome, systemic manifestations, and increased risk of lymphoma [1]. As such, it has a broadened spectrum of clinical manifestations ranging from isolated exocrinopathy or widespread chronic pain syndrome to life-threatening systemic manifestations and lymphoma development. Because of this large variability in its clinical manifestations, relentless efforts have been made to delineate patients having a higher likelihood of a worsened prognosis. In this context, several biomarkers have been studied to define subgroups of patients with pSS with a worsened prognosis [2].

The discovery of new autoantibodies has been, up until now, only botched attempts [3]. Currently, the diagnostic and prognostic biomarkers used in daily clinical practice still remain the conventional immunological serum markers, as evidence of an underlying systemic autoimmune process: antinuclear antibodies (ANA), rheumatoid factor (RF), hypergammaglobulinemia, cryoglobulinemia, and decreased serum complement C4. The notion of “seropositivity” in pSS is complex because, classically, it was based on the serological item of consensus classification criteria. In the 1986 San Diego and 1993 European criteria for the diagnosis of pSS, the serological criteria corresponded to the positivity of at least one marker between (1) antibodies to Ro/SSA or La/SSB antigens, (2) antinuclear antibodies, and/or (3) the rheumatoid factor. In the 2002 American-European classification criteria for pSS, only the positivity of an anti-Ro/SSA and/or anti-La/SSB antibody was retained [4]. In the 2012 American College of Rheumatology (ACR)/SICCA classification criteria, the serological item corresponded to (1) serum positivity of anti-Ro/SSA and/or anti-La/SSB or (2) a positive rheumatoid factor with an antinuclear antibody titer >1:320 [5]. Finally, in the more recent 2016 ACR-European Alliance of Associations for Rheumatology (EULAR) classification criteria, only anti-Ro/SSA antibody positivity remained [6].

In addition to its pivotal role as a diagnostic biomarker, anti-Ro/SSA positivity, present in 70–80% of patients with pSS, is associated with the serological profile, systemic extraglandular manifestations of the disease, and lymphoma. Patients are statistically younger, with predominating systemic autoimmune manifestations and a lower frequency of sicca syndrome [7,8,9]. Conversely, seronegative patients are older, classically characterized by sicca syndrome and chronic pain. This seronegative subgroup of patients with pSS is portrayed by dampened systemic and inflammatory features and does not carry an increased risk of developing lymphoma [7,8,9]. Therefore, autoantibodies directed at the Ro/La antigens complex play a central role in the diagnosis and prognosis of pSS. As an ANA-associated disease, the study of the serological profiles of patients with pSS differentiates “classical ANA”-positive pSS (dominated by an anti-Ro/SSA-driven phenotype) and “atypical ANA”-positive pSS (mainly involving anti-centromere-positive pSS). Non-identified ANA are defined by the presence of ANA but not specified as either SSA or SSB or any other extractable nuclear antigen (ENA) specificity, including anti-centromere or anti-Smith (Sm). However, to our knowledge, the clinical phenotype of patients with pSS with non-identified ANA remains poorly known.

The aim of this study was to characterize retrospectively the clinical phenotype of patients with pSS with a non-identified ANA profile and no other concurrent autoimmune rheumatic diseases (ARD), in a comparison with that of patients with pSS with negative ANA, positive typical ANA (anti-Ro/SSA and/or La/SSB), and positive atypical ANA. Notably, this is, to our knowledge, the first study that sought to describe the clinical features of patients with pSS with non-identified ANA profiles and highlight their distinctiveness from other ANA subgroups.

## 2. Materials and Methods

We conducted an observational, retrospective monocentric study at the Erasme University Hospital (Brussels, Belgium) on 233 patients with pSS. The patient database was constructed by cross-referring all the patients having a positive salivary gland biopsy as well as the patients with positive anti-SSA/SSB antibodies. All the patients fulfilling the 2002 American–European Consensus Group (AECG) criteria for pSS, considering the exclusion criteria, were included in this study. The patients with other associated systemic diseases were excluded.

The following data were systematically extracted from the computerized medical records of the selected patients: (I) demographic data: gender, age at diagnosis (date of biopsy), and duration of follow-up in our institution; (II) sicca asthenia polyalgia syndrome: fatigue, chronic pain, dry eye, and dry mouth symptoms, Schirmer’s test, salivary gland scintigraphy; (III) systemic activity assessed by ESSDAI items and total score assessed retrospectively; (IV) accessory salivary gland biopsy results expressed as Chisholm and Mason scores; (V) laboratory data concerning ANA profiles tested by indirect immunofluorescence tests on HEp-2 cells and subsequent identification by ELISA, others systemic or organ-specific autoantibodies, cryoglobulinemia, C4 consumption, RF, and hypergammaglobulinemia; and (VI) treatments used, e.g., corticosteroids, hydroxychloroquine, and other immunosuppressant.

Data analyses were performed using SPSS Statistics version 13 (IBM, Chicago, IL, USA) and GraphPad Prism 8 for Windows (GraphPad Software, San Diego, CA, USA). The normality of the distributions was determined using the Shapiro–Wilk test. Standard descriptive statistics were used, including percentages, proportions, median, and range. Fisher’s exact probability test (FEPT) and Chi-square test (CST) were used for the contingency tables (qualitative or dichotomic data), and the Kruskal–Wallis test with a post hoc Dunn’s test (KWT) was used for continuous data. A *p*-value < 0.05 was considered statistically significant.

To explore the simultaneous relationships between the variables, multiple correspondence analysis (MCA) was applied. This technique allows the exploration of the relationships between subgroups of interest together with the other exploratory variables by the comparison of distances and clustering in a multiple-dimension space. The MCA analysis was generated using the SAS software (2020, SAS Institute Inc., Cary, NC, USA).

## 3. Results

A total of 233 patients who fulfilled the AECG classification criteria were included (Table 1). They were predominantly women (93.6%), with a median age at diagnosis of 52 years (range 17–85). A minor salivary gland biopsy was performed in 209 patients (89.7% of the cohort), and focal sialadenitis with a Chisholm’s score ≥ 3 was found in 177/209 of the patients (85%). ANA positivity at ≥1:80 and ≥1:320 was 74.2% and 40.3%, respectively. A total of 42 of the 233 patients (18%) had non-identified ANA. Anti-Ro/SSA and anti-La/SSB antibodies were found in 124/233 of the patients (53.2%) and 52/233 of the patients (22.4%), respectively. The median ESSDAI at diagnosis—calculated retrospectively—was 2 (range 0–23), globally reflecting a low systemic activity but with a number of patients presenting with severe systemic manifestations.

The anti-Ro/SSA-positive patients were significantly younger at diagnosis compared to the ANA-negative patients (KWT, *p* ≤ 0.001), and their ESSDAI score at diagnosis was statistically higher compared to the ANA-negative (KWT, *p* ≤ 0.001) and non-identified ANA patients (KWT, *p* ≤ 0.01) (Figure 1A,B, respectively). No statistically significant difference could be observed between the non-identified ANA patients and the ANA-negative patients in terms of age and/or ESSDAI score at diagnosis.

There were more frequent articular manifestations, positive RF, and use of corticosteroids in the anti-Ro/SSA-positive patients compared to the ANA-negative (FEPT, *p* ≤ 0.0001 for all three parameters) and non-identified ANA patients (FEPT, *p* ≤ 0.01 for all three parameters) (Figure 2A–C). On the other hand, a statistically significant difference was established in terms of positive RF and corticosteroid treatment in the non-identified ANA-positive patients compared to the ANA-negative patients (FEPT, *p* ≤ 0.05 for the two parameters). There were more frequent articular manifestations in the non-identified ANA compared to ANA-negative patients, but no statistical significance was reached (12/42 vs. 9/60, FEPT, *p* = 0.135).

A total of 9 out of 120 anti-Ro/SSA-positive patients were also positive for other atypical antibodies such as anti-centromere (*n* = 1), low levels of anti-dsDNA (*n* = 5), anti-nucleosome (*n* = 3), anti-U1RNP (*n* = 1), anti-Sm (*n* = 1), anti-Mi2 (*n* = 1), and anti-PM/Scl75 + anti-PL7 double positivity (*n* = 1). However, these patients did not meet the diagnostic criteria for any concomitant ARD, including systemic lupus erythematosus (SLE), systemic sclerosis (SSc), or mixed connective tissue disease (MCTD).

Peripheral nervous system manifestations (PNS) were statistically more frequent in the non-identified ANA and ANA-negative patients compared to the anti-Ro/SSA-positive patients (FEPT, *p* ≤ 0.01 and *p* ≤ 0.001, respectively) (Figure 2D). There was no statistically significant difference for PNS manifestations between the non-identified ANA and ANA-negative patients (7/41 vs. 11/59, FEPT, *p* = 1).

Next, we explored the relationships between serological ANA profiles and clinical variables by multiple correspondence analysis. Three dimensions were selected for this analysis, explaining 44.8% of data variability. The use of a greater number of dimensions made the analysis difficult to interpret. In this analysis, dimension 1 opposed the ANA-negative and anti-SSA-positive patients with pSS, dimension 2 opposed the anti-SSA-positive and atypical ANA-positive pSS, and dimension 3 opposed the ANA-negative and atypical ANA-positive patients with pSS.

In Figure 3, the anti-SSA-positive pSS profile corresponds to the lower right quadrant. The anti-SSA-positive pSS profile is characterized more frequently by fatigue, articular manifestations, biological manifestations, hypergammaglobulinemia, and hydroxychloroquine use. PNS involvement is less frequent in the anti-SSA-positive patients with pSS. Moderate-to-severe ESSDAI, the use of immunosuppressant and/or methylprednisolone, rheumatoid factor positivity, hypergammaglobulinemia, and a lower prevalence of xerostomia are depicted by the anti-SSA-positive and atypical ANA-positive profiles, relative to the ANA-negative and non-identified ANA-positive profiles.

In Figure 4, the ANA-negative pSS profile corresponds to the lower left quadrant. These patients are essentially characterized not only by a less frequent use of immunosuppressive therapy and/or methylprednisolone and by the paucity of joint manifestations but also by a higher proportion of patients having an ESSDAI score of zero. In this figure, the non-identified ANA-positive status differs from the ANA-negative profile, reflecting two distinct profiles.

In Figure 5, the atypical ANA-positive pSS profile corresponds to the upper right quadrant. These patients are distinguished by a higher frequency of PNS involvement, moderate-to-severe ESSDAI, and the use of immunosuppressive and/or methylprednisolone therapy, relative to the other pSS subsets.

## 4. Discussion

For the first time to our knowledge, our study has characterized the clinical phenotype of patients with pSS with non-identified ANA at diagnosis. In previous studies, ANA positivity was generally used as a biomarker regardless of its identification, or the non-identified ANA-positive subgroup of patients with pSS was excluded [10,11]. For instance, Fossaluza et al. assessed the clinical differences of patients with primary Sjögren’s syndrome based on either ANA and/or anti-ENA antibody-negative or -positive cases, therefore excluding the non-identified ANA-positive subgroup [11]. On the other hand, Chatzis et al. compared the clinical and biological profiles of patients with different serological profiles, including triple seronegativity (ANA(+) but negative for anti-Ro/SSA, anti-La/SSB, and RF) and quadruple seronegativity (negative for ANA, anti-Ro/SSA, anti-La/SSB, and RF). However, they did not investigate whether the triple seronegativity group had other ENA specificities and excluded non-identified ANA-positive patients with positive RF+. Overall, they showed that triple-seronegative patients had a higher incidence of persistent lymphadenopathy and lymphoma, a higher focus score, and a later age of SS diagnosis compared to quadruple-seronegative patients [10].

Our cohort of 233 patients with primary Sjögren’s syndrome confirmed the demographical, clinical, and serological characteristics that had been previously reported in larger cohorts [12,13,14,15]. Our 14:1 female-to-male ratio is in agreement with the female-to-male ratio in previous cohorts, ranging from 8:1 [14,16,17] to 20:1 [15,18]. The mean age at diagnosis was 52 years old in our study group and is comparable to other cohort studies [1,19]. In line with findings previously reported in the literature, we found that 173 patients (74%) were positive for ANA and 124 patients (53%) presented with anti-Ro/SSA positivity, respectively. Indeed, ANA positivity ranged from 74 to 94% and anti-Ro/SSA autoantibody positivity from 40 to 68% in the largest cohorts reported in the literature [19,20,21].

In our cohort, 42 patients (18%) presented a non-identified ANA-positive profile. The distribution of ANA titers observed in this subgroup was different from that observed in “healthy controls” previously published [22] and, therefore, cannot be considered as a non-specific variant of the normal. Overall, these patients have a clinical phenotype sharing several distinguished features resembling those depicted by ANA-negative patients. Nevertheless, we observed that certain distinctive features were encompassed by the non-identified ANA-positive group, such as a tendency towards a younger age at diagnosis, more frequent RF positivity, and a higher proportion of corticosteroid use compared to the ANA-negative patients. These differences certainly reflected a greater systemic activity assessed by the attending physician, given that the ESSDAI score at diagnosis was not statistically different between these two groups. Since the inflammatory joint manifestations were the dominant ESSDAI item in this subset, it is probable that patients with non-identified ANA have greater inflammatory joint involvement then ANA-negative patients. Patients with non-identified ANA have a significantly higher proportion of RF positivity compared to ANA-negative patients but a significantly lower proportion than anti-Ro/SSA-positive patients. This significantly increased proportion of RF positivity may reflect an increase in articular manifestations in this group of patients. However, this hypothesis will need to be validated in other patient cohorts.

In our cohort, 124 patients (53.2%) presented anti-Ro/SSA positivity, and 51 patients (21.8%) presented anti-Ro/SSA and anti-La/SSB double positivity. Anti-SSA-positive patients are statistically younger at diagnosis and report more fatigue but less xerostomia than anti-SSA-negative patients. The anti-SSA-positive subgroup of patients in our study showed systemic manifestations. In our cohort, glandular, lymphadenopathy, cutaneous, pulmonary and hematologic involvement were found exclusively in the anti-SSA-positive patients. Inflammatory joint and biological involvement are more common in patients with anti-Ro/SSA antibodies than in those without such antibodies. Almost half of anti-SSA-positive patients have hypergammaglobulinemia and/or RF positivity. In the latter subgroup, in our study, immunosuppressive and/or corticosteroid therapy were most frequently used. These results confirm the major prognostic role of anti-Ro/SSA positivity in patients with pSS.

In our cohort, only one patient had anti-Ro/SSA and anti-centromere double positivity. This rare “hybrid” association has already been studied by Suzuki et al. in a cohort of 108 patients with pSS. In this study, the patients with pSS with anti-centromere/SSA double positivity (16/108, 14.8%) were significantly older than the anti-SSA-positive patients but had a higher ESSDAI at diagnosis than the anti-centromere patients with pSS. These double-positive patients also had the highest prevalence of Raynaud’s phenomenon (56.3%) compared to the other groups. Although this cohort was created from all the biopsies performed in a single center, we cannot explain the rarity of this manifestation in our cohort compared to the high prevalence of double positivity reported in this study.

Only seven patients (3%) presented atypical ANA positivity in our pSS cohort. Due to this small number of patients, it was not feasible to perform statistical comparisons with this subgroup. In essence, patients with atypical ANA have more salivary gland enlargement, Raynaud phenomena, number of systemic involvement, hypergammaglobulinemia, and use of corticosteroid and immunosuppressive drugs [23]. Our multiple correspondence analysis confirmed this association by demonstrating the colocalization of these manifestations in the quadrant corresponding to the atypical ANA-positive profile (Figure 5). A 49-year-old patient presented with sicca syndrome associated with isolated anti-La/SSB positivity, found on at least three occasions over 5 years, and a Chisholm score of 4 on their minor salivary gland biopsy. This patient had no systemic manifestation at diagnosis (ESSDAI = 0). This profile, found in 2–3% of pSS cohorts, has already been individualized in the past. Indeed, in the original study by Baer et al. [24] based on patients with pSS according to the SICCA 2012 criteria, the phenotype of these patients did not differ from that of seronegative patients. However, data from cohorts of patients with pSS, classified as such according to the AECG 2002 criteria, showed that this small subgroup of patients could feature a less severe phenotype than SSA-positive patients, characterized by a more marked sicca syndrome compared to anti-SSA-positive patients but more systemic manifestations than seronegative patients [25]. Four (1.7%) patients in our cohort belonged to the anti-centromere subset described in the literature. This prevalence is lower than that of 3.7–27% reported in other studies [26]. This phenotype is classically associated with an older age at diagnosis, a lower positivity for anti-Ro/SSA, RF, and hypergammaglobulinemia. These patients present more Raynaud phenomena or even more CREST-like manifestations. Finally, the last two patients presented, respectively, anti-RNP and anti-Ku antibodies. The first one was a 23-year-old female patient with pSS-associated sicca syndrome, neuronopathy, and polyneuropathy, and the other was a 40-year-old female patient with sicca syndrome and non-specific arthro-myalgia but not myositis.

We also reported an increased prevalence of PNS involvement, according to the ESSDAI item, in the seronegative and non-identified ANA-positive patients compared to the anti-SSA-positive patients. These data are in agreement with those of a retrospective cohort of 420 patients with pSS according to the AECG 2002 criteria, out of which 62 patients had peripheral neurological manifestations (14.8%) [27]. In this study, the prevalence of anti-Ro/SSA antibodies was lower in the affected patients than in the non-affected patients. Although not included in the ESSDAI, pSS-associated sensory small-fiber neuropathies have also been associated with a lower prevalence of anti-Ro/SSA positivity in a prospective series of 40 cases, compared to 100 unaffected patients with pSS [28]. In a retrospective cohort of 120 patients with pSS, Sene et al. [28] reported sensorimotor neuropathy (SMN) or nonataxic sensory neuropathy (SN) in 7 and 20 patients, respectively. The SMN group was characterized by an increased prevalence of cryoglobulinemia. The SN group was associated with a lower prevalence of ANA and anti-Ro/SSA positivity. The heterogeneity in the pathophysiology of peripheral neurological involvement in pSS and the cases’ definition could explain the conflicting results that were published [17]. Scofield et al. [29] reported 27 (31%) cases of clinically defined polyneuropathies in a cohort of 88 patients. In their study, sensory peripheral neuropathy was associated with the presence of anti-Ro/SSA positivity. In our cohort, the proportion of patients with PNS involvement was lower than that in these studies (10.8%). This apparent discrepancy may be related to the assessment of PNS at the time of diagnosis in our cohort and not in the longitudinal follow-up of the patients. Polyneuropathies have been shown to be an incidental systemic manifestation commonly reported in the months following a diagnosis [30].

The limitations of our study lie with the retrospective nature of this study, based on the review of medical files, the exhaustiveness of which was left to the discretion of the attending physician, and the retrospective calculation of the ESSDAI based on the available data, which could have led to its underestimation. In addition, the small number of patients included for statistical analysis, especially in the ANA+ atypical subgroup (*n* = 7), may also have reduced the statistical power of this study, potentially limiting the generalizability of our findings.

## 5. Conclusions

For the first time to our knowledge, our study characterized the clinical phenotype of patients with pSS with non-identified ANA at diagnosis. The non-identified ANA-positive patients were characterized by a clinical phenotype similar to the ANA-negative patients but with less systemic and biological activity and with more PNS manifestations than the anti-Ro/SSA-positive patients. Furthermore, the non-identified ANA-positive patients were mainly distinguished from the ANA-negative patients by a greater proportion of RF positivity and the need for corticosteroid use due to a higher frequency of articular involvement.

## Figures and Tables

**Figure 1 diagnostics-14-00935-f001:**
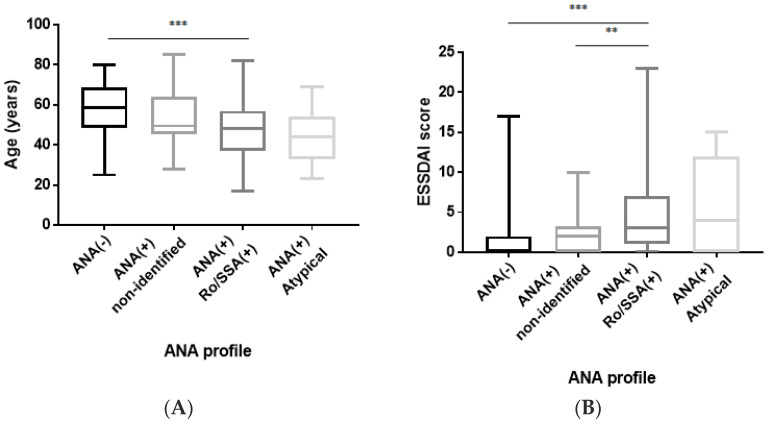
(**A**) Comparison of age at diagnosis according to the ANA profile. Anti-Ro/SSA-positive patients are statistically younger at diagnosis than ANA-negative patients (KWT, *p* ≤ 0.001). Data are depicted as a box-and-whisker plot. *** *p* < 0.001. ANA: antinuclear antibody. (**B**) ESSDAI score at diagnosis according to the ANA profile. The ESSDAI score is statistically higher at diagnosis in anti-Ro/SSA-positive patients compared to ANA-negative (KWT, *p* ≤ 0.001) and non-identified ANA (KWT, *p* ≤ 0.01) patients. Data are depicted as a box-and-whisker plot. ** *p* < 0.01; and *** *p* < 0.001. ANA: antinuclear antibody; and ESSDAI: EULAR Sjögren’s syndrome disease activity index.

**Figure 2 diagnostics-14-00935-f002:**
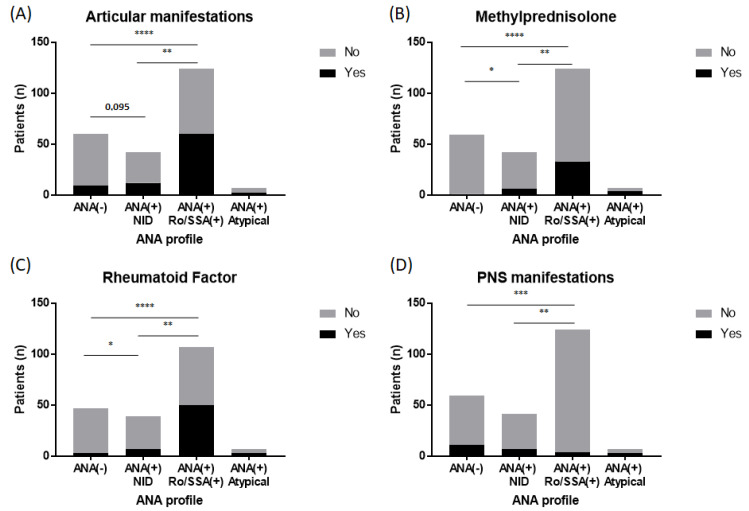
ANA profile according to clinical, biological, and therapeutic characteristics. (**A**) Articular manifestations are statistically more frequent at diagnosis in anti-Ro/SSA-positive patients compared to ANA-negative (FEPT *p* ≤ 0.0001) patients and non-identified ANA patients (FEPT *p* ≤ 0.01). (**B**) Methylprednisolone is more frequently prescribed at diagnosis in anti-Ro/SSA-positive patients compared to ANA-negative (FEPT *p* ≤ 0.0001) patients and non-identified ANA patients (FEPT *p* ≤ 0.01). It is also more frequently prescribed at diagnosis in non-identified ANA patients compared to ANA-negative patients (FEPT *p* ≤ 0.05). (**C**) A positive rheumatoid factor at diagnosis is more frequent in anti-Ro/SSA-positive patients compared to ANA-negative (FEPT *p* ≤ 0.0001) patients and non-identified ANA patients (FEPT *p* ≤ 0.01). It is also more frequent at diagnosis in non-identified ANA patients compared to ANA-negative patients (FEPT *p* ≤ 0.05). (**D**) Peripheral nervous system manifestations are statistically less frequent at diagnosis in anti-Ro/SSA-positive patients compared to ANA-negative (FEPT *p* ≤ 0.001) patients and non-identified ANA patients (FEPT *p* ≤ 0.01). Data are expressed as the number of cases. * *p* < 0.05; ** *p* < 0.01; *** *p* < 0.001; and **** *p* < 0.0001. ANA: antinuclear antibody; and PNS: peripheral nervous system.

**Figure 3 diagnostics-14-00935-f003:**
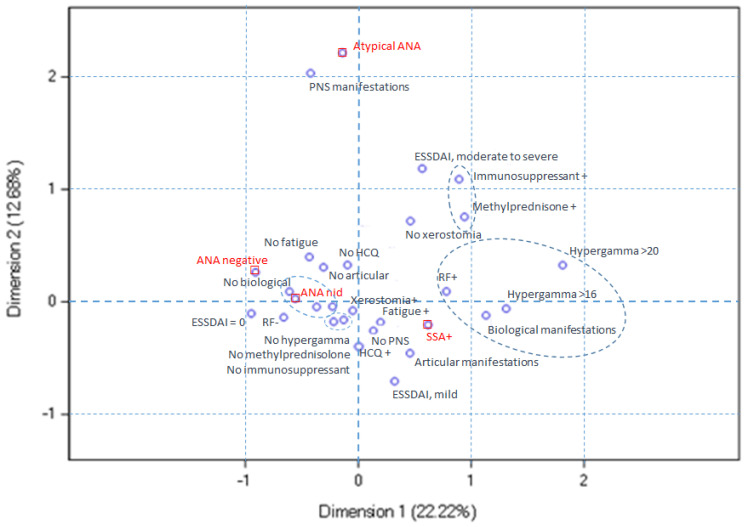
Multiple correspondence analysis plot for dimensions 1 and 2. The anti-SSA-positive pSS profile, located in the lower right quadrant, is characterized by more frequent fatigue, articular and biological manifestations, hypergammaglobulinemia, and hydroxychloroquine use. PNS involvement is less common in these patients. Both the anti-SSA-positive and atypical ANA-positive profiles exhibit moderate-to-severe ESSDAI scores, immunosuppressant and/or methylprednisolone use, positive rheumatoid factor, hypergammaglobulinemia, and reduced xerostomia prevalence compared to the ANA-negative and non-identified ANA-positive profiles.

**Figure 4 diagnostics-14-00935-f004:**
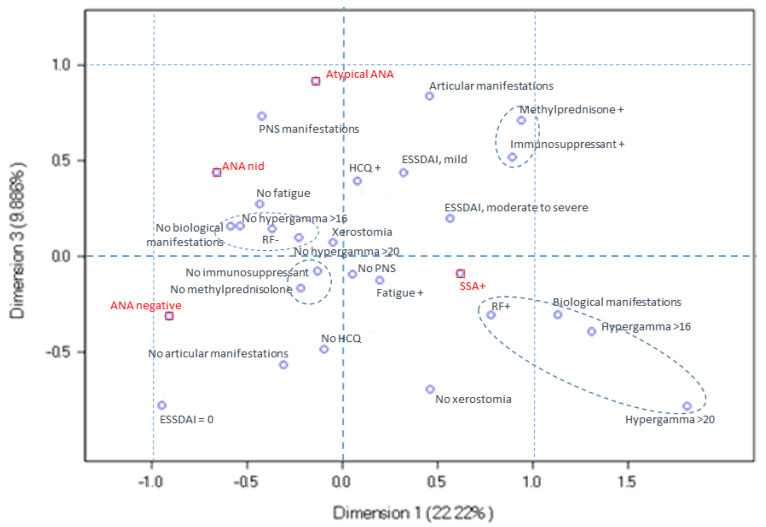
Multiple correspondence analysis plot for dimensions 1 and 3. The ANA-negative pSS profile, situated in the lower left quadrant, exhibits the infrequent utilization of immunosuppressive therapy and/or methylprednisolone, diminished joint manifestations, and a heightened prevalence of patients with an ESSDAI score of zero. The non-identified ANA-positive status differs from the ANA-negative profile, reflecting two distinct profiles.

**Figure 5 diagnostics-14-00935-f005:**
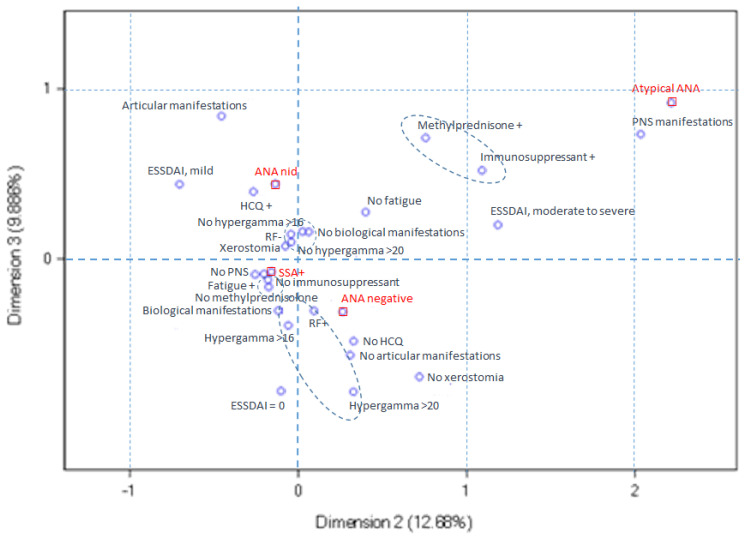
Multiple correspondence analysis ploy for dimensions 2 and 3. The atypical ANA-positive pSS profile, located in the upper right quadrant, is characterized by an elevated incidence of peripheral nervous system (PNS) involvement, moderate-to-severe ESSDAI scores, and the utilization of immunosuppressive and/or methylprednisolone therapy compared to the other pSS subsets.

**Table 1 diagnostics-14-00935-t001:** Clinical and biological data of the patients with pSS with different serological profiles of autoantibodies. The different subgroups of patients with pSS are compared for their clinical, biological, and therapeutic characteristics. All the values are *n* (%) or the median [range]. Row data have been compared using the Kruskal–Wallis test for the continuous variables and Fisher’s exact probability test or Chi-square test for discontinuous the variables.

		pSS Cohort	ANA Negative	ANA (+) Non-Identified	ANA (+) Anti-Ro/SSA	ANA (+) Atypical	*p*-Value
		(*n* = 233)	(*n* = 60)	(*n* = 42)	(*n* = 124)	(*n* = 7)	
DEMOGRAPHIC DATA							
Female	*n* (%)	218 (93.6)	55 (91.7)	42 (100)	114 (91.9)	7 (100)	0.233
Age at diagnosis	Med	52 [17–85]	58.5 [25–80]	49.5 [28–85]	48 [17–82]	44 [23–69]	0.0009
SICCA ASTHENIA POLYALGIA COMPLEX							
Fatigue	*n* (%)	156/229 (68)	31 (51.7)	26 (61.9)	96/120 (80)	3 (42.9)	<0.0001
Chronic pain	*n* (%)	171/232 (74)	41 (68.3)	31 (73.8)	95/123 (77.2)	4 (57.1)	0.444
Dry eyes	*n* (%)	212 (90.6)	56 (93.3)	41 (97.6)	108 (87.1)	6 (85.7)	0.178
Xerostomia	*n* (%)	210 (90.1)	59 (98.3)	40 (95.2)	105 (84.7)	6 (85.7)	0.009
Positive Schirmer’s test	*n* (%)	93/146 (63.7)	25/40 (62.5)	19/25 (76)	46/76 (60.5)	3/5 (60)	0.568
Positive SG Scintigraphy	*n* (%)	37/66 (56.1)	4/10 (40)	4/7 (57.1)	29/47 (61.7)	0/2 (0)	0.240
SYSTEMIC MANIFESTATIONS							
Constitutional	*n* (%)	5 (2.1)	0 (0)	1 (2.4)	4 (3.2)	0 (0)	_
Lymphadenopathy	*n* (%)	8/231 (3.5)	0 (0)	0 (0)	8/123 (6.5)	0 (0)	_
Glandular	*n* (%)	7 (3)	0 (0)	0 (0)	7 (5.6)	0 (0)	_
Articular	*n* (%)	83 (35.6)	9 (15)	12 (28.6)	60 (48.4)	2 (28.6)	<0.0001
Cutaneous	*n* (%)	11 (4.7)	0 (0)	0 (0)	11 (8.9)	0 (0)	_
Pulmonary	*n* (%)	14/232 (6)	0 (0)	0 (0)	14/123 (11.4)	0 (0)	_
Renal	*n* (%)	4 (1.7)	0 (0)	0 (0)	4 (3.2)	0 (0)	_
Muscular	*n* (%)	1 (0.4)	0 (0)	(0)	1 (0.8)	0 (0)	_
PNS	*n* (%)	25/231 (10.8)	11/59 (18.6)	7/41 (17.1)	4 (3.2)	3 (42.9)	<0.0001
CNS	*n* (%)	9 (3.9)	4 (6.7)	1 (2.4)	4 (3.2)	0 (0)	_
Hematological	*n* (%)	3/232 (1.3)	0 (0)	0 (0)	3 (2.4)	0 (0)	_
Biological	*n* (%)	76 (32.6)	6 (10)	4 (9.5)	65 (52.4)	1 (14.3)	<0.0001
ESSDAI at diagnosis	Med	2 [0–23]	0 [0–17]	2 [0–10]	3 [0–23]	4 [0–15]	<0.0001
ESSDAI = 0	*n* (%)	80 (34.3)	36 (60)	19 (45.2)	23 (18.5)	2 (28.6)	<0.0001
ESSDAI < 5	*n* (%)	88 (37.8)	11 (18.3)	14 (33.3)	61 (49.2)	2 (28.6)	
ESSDAI ≥ 5	*n* (%)	65 (27.9)	13 (21.7)	9 (21.4)	40 (31.3)	3 (42.9)	
LABORATORY DATA							
Chisholm Score							
Chisholm 0	*n* (%)	14/209 (6.7)	0 (0)	0 (0)	14/100 (14)	0 (0)	_
Chisholm 1–2	*n* (%)	18/209 (8.6)	0 (0)	0 (0)	18/100 (18)	0 (0)	_
Chisholm 3–4 (FC ≥ 1)	*n* (%)	177/209 (85)	60 (100)	42 (100)	68/100 (68)	7 (100)	_
ANA pattern							
Speckled	*n* (%)	152 (65)	0 (0)	28 (66.7)	121 (97.6)	3 (42.9)	_
Homogeneous	*n* (%)	26 (11.2)	0 (0)	18 (42.9)	7 (5.6)	1 (14.3)	_
Nucleolar	*n* (%)	13 (5.6)	0 (0)	5 (11.9)	8 (6.5)	0 (0)	_
Others	*n* (%)	8 (3.2)	0 (0)	2 (4.8)	2 (1.6)	4 (57.1)	_
ANA titer							
1:80	*n* (%)	51 (25.8)	0 (0)	17 (40.5)	34 (27.4)	0 (0)	_
1:160	*n* (%)	28 (12)	0 (0)	8 (19)	20 (16.1)	0 (0)	_
1:320	*n* (%)	20 (8.6)	0 (0)	7 (16.7)	12 (9.2)	1 (14.3)	_
1:640	*n* (%)	24 (10.3)	0 (0)	2 (4.8)	21 (16.9)	1 (14.3)	_
≥1:1280	*n* (%)	50 (21.4)	0 (0)	8 (19)	37 (29.8)	5 (71.4)	_
ANA identification							
Anti-Ro/SSA	*n* (%)	124 (53.2)	0 (0)	0 (0)	124 (100)	0 (0)	_
Anti-La/SSB	*n* (%)	52 (22.4)	0 (0)	0 (0)	51 (41.1)	1 (14.3)	_
Others ANA	*n* (%)	15/227 (6.6)	0 (0)	0 (0)	9/120 (7.5)	6 (85.7)	_
Others immunological markers							
Cryoglobulinemia	*n* (%)	16/138 (11.6)	3/44 (6.8)	3/31 (9.7)	9/57 (15.8)	1/6 (16.7)	0.528
C4 consumption	*n* (%)	11/161 (6.8)	1/35 (2.9)	0/28 (0)	10/96 (10.4)	0/2 (0)	0.171
γ-globulins > 1600 g/dL	*n* (%)	65/231 (28.1)	3 (5)	2 (4.5)	59/122 (48.4)	1 (14.3)	<0.0001
γ-globulins > 2000 g/dL	*n* (%)	23/231 (10)	1 (1.7)	0 (0)	22/122 (18)	0 (0)	<0.0001
Rheumatoid Factor	*n* (%)	63/200 (31.5)	3/47 (6.4)	7/39 (17.9)	50/107 (46.7)	3 (42.9)	<0.0001
THERAPEUTICS							
Hydroxychloroquine	*n* (%)	122/231 (53)	34/58 (58.6)	27/42 (64.3)	60 (48.4)	1 (14.3)	0.054
Immunosuppressant	*n* (%)	26 (11.2)	3 (5)	1 (2.4)	19 (15.3)	3 (42.9)	0.002
Methylprednisone PO	*n* (%)	44/232 (19)	1/59 (1.7)	6/42 (14.3)	33 (26.6)	4 (57.1)	<0.0001

pSS: primary Sjögren’s syndrome; SG: salivary glands; PNS: peripheral nervous system; CNS: central nervous system; ESSDAI: EULAR Sjögren’s syndrome disease activity index; ANA: antinuclear antibody; and PO: per os.

## Data Availability

The data used for this study are not available due to ethical and privacy restrictions.

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
