# Peer review of "Clinical Profile of Patients with Primary Sjögren’s Syndrome with Non-Identified Antinuclear Autoantibodies"

_diagnostics, 2024, doi:10.3390/diagnostics14090935_

Round 1

Reviewer 1 Report

Comments and Suggestions for Authors

The authors present a retrospective cross-sectional study of the clinical aspects of Sjogren's syndrome with unidentified antinuclear antibodies in serum.

Remarks:

1)      The definition of the acronym EULAR (European Alliance of Associations for Rheumatology) should be updated.

2)      In the introduction, the authors should better clarify that the patients with unidentified ANA were those ANA+ but negative for anti-SSA/anti-SSB and negative for other ENA.

3)      The authors should specify why some patients were treated with corticosteroids or immunosuppressants. Was Sjogren's syndrome associated with SLE or rheumatoid arthritis?

4)      The authors should specify whether the presence of other autoimmune diseases (SLE, mixed connective tissue disease, systemic sclerosis, etc.) were excluded in the nine patients positive for atypical autoantibodies (anti-centromere, anti-dsDNA, etc.).

Comments on the Quality of English Language

English needs only minor editing

Author Response

Reviewer 1

The authors present a retrospective cross-sectional study of the clinical aspects of Sjogren's syndrome with unidentified antinuclear antibodies in serum.

Remarks:

  • The definition of the acronym EULAR (European Alliance of Associations for Rheumatology) should be updated.

It has been updated

  • In the introduction, the authors should better clarify that the patients with unidentified ANA were those ANA+ but negative for anti-SSA/anti-SSB and negative for other ENA.

It has been clarified

  • The authors should specify why some patients were treated with corticosteroids or immunosuppressants. Was Sjogren's syndrome associated with SLE or rheumatoid arthritis?

In the materials and methods, it is specified that ‘Patients with other associated systemic disease were excluded from our study.” Patients were treated with corticosteroids or immunosuppressants to tackle symptoms related to their pSS.

To avoid any confusion, we edited the intro to clarify the definition of the included patients in our study.

  • The authors should specify whether the presence of other autoimmune diseases (SLE, mixed connective tissue disease, systemic sclerosis, etc.) were excluded in the nine patients positive for atypical autoantibodies (anti-centromere, anti-dsDNA, etc

We have clarified it by stating that these 9 patients didn’t meet criteria for any other autoimmune rheumatic diseases including SLE, MCTD, SSc, etc.

Reviewer 2 Report

Comments and Suggestions for Authors

The topic is interesting and the paper is quite well written. However, I think that some parts need to be improved, I have some comments:

1) Abstract. Results - In our cohort, 42 patients (18%) presented a non-identified ANA-positive profile. Anti-Ro/SSA positive patients were significantly younger at diagnosis compared to ANA-negative patients (p≤0.001) and their ESSDAI score at diagnosis was statistically higher compared to ANA-negative (p≤0.001) and nonidentified ANA patients (p≤0.01). There were significantly more frequent articular manifestations, positive Rheumatoid Factor (RF) and use of corticosteroids in anti-Ro/SSA positive-patients compared to ANAnegative (p≤0.0001) and non-identified ANA-positive patients (p≤0.01). A significantly higher proportion of RF positivity and corticosteroids treatment was observed in non-identified ANA-positive patients compared to ANA-negative patients (p<0.05). Pleease, underline the most important results to support the conclusions.

2) Conclusion - The non-identified ANA-positive patients featured a clinical phenotype similar to ANA-negative patients. On the other hand, non-identified ANA-positive patients were mainly distinguished from ANAnegative patients by a greater proportion of RF positivity and need of corticosteroid use due to articular involvement. Abstract might be beneficial to include a sentence that briefly summarizes the topic  of the study and to underline the novelty of the study.

3) Introduction. The aim of the study was to characterize retrospectively the clinical phenotype of pSS patients with nonidentified ANA profile, in comparison with that of pSS patients with negative ANA, positive typical ANA (anti-Ro/SSA and/or La/SSB) and positive atypical ANA. Please, improve the description of study aim and the underline the novelty of these observations.

4) Data analysis were performed using SPSS Statistics version 13 (IBM, Chicago, Illinois, USA) and GraphPad Prism 8 for Windows (GraphPad Software, San Diego, California, USA). Standard descriptive statistics were used, including percentages, proportions, median and range. Fisher’s exact probability test and Chi-square test were used for contingency tables (qualitative date) and Kruskal–Wallis test was used for continuous data. A p-value < 0.05 was considered statistically significant. To explore the simultaneous relationships between variables, multiple correspondence analysis (MCA) was applied. This technique allows the exploration of the relationships between sub-groups of interest together with the other exploratory variables by comparison of distances and clustering in a multiple dimension space. MCA analysis was generated using SAS software (2020, SAS Institute Inc., Cary, NC, USA). Please, add some information on statistical tests used to evaluate the data and regarding data presentation.

5) Results. Please, improve the description of the figure.

6) Discussion For the first time to our knowledge, our study has characterized the clinical phenotype of pSS patients with non-identified ANA at diagnosis. In previous studies, ANA positivity was generally used as biomarker regardless of its identification, or the non-identified ANA positive subgroup of pSS patients was excluded (10, 11). The discussion section needs to be improved.  It is necessary improve the comparison with published literature.

7) The limitations of our study lie with the retrospective nature of the study based on the review of medical files and the exhaustiveness of which was left to the discretion of the attending physician; the retrospective calculation of ESSDAI based on the available data, which could lead to its underestimation; and the small numbers of patients in certain subgroups. Please ameliorate this paragraph I also suggest including the number of patients evaluated for the final analysis as a limit

Comments on the Quality of English Language

Minor changes of English language are required

Author Response

Reviewer 2

The topic is interesting and the paper is quite well written. However, I think that some parts need to be improved, I have some comments:

  • Results –

In our cohort, 42 patients (18%) presented a non-identified ANA-positive profile. Anti-Ro/SSA positive patients were significantly younger at diagnosis compared to ANA-negative patients (p≤0.001) and their ESSDAI score at diagnosis was statistically higher compared to ANA-negative (p≤0.001) and nonidentified ANA patients (p≤0.01). There were significantly more frequent articular manifestations, positive Rheumatoid Factor (RF) and use of corticosteroids in anti-Ro/SSA positive-patients compared to ANAnegative (p≤0.0001) and non-identified ANA-positive patients (p≤0.01). A significantly higher proportion of RF positivity and corticosteroids treatment was observed in non-identified ANA-positive patients compared to ANA-negative patients (p<0.05).

Please, underline the most important results to support the conclusions.

It has been underlined

  • Conclusion –

The non-identified ANA-positive patients featured a clinical phenotype similar to ANA-negative patients. On the other hand, non-identified ANA-positive patients were mainly distinguished from ANAnegative patients by a greater proportion of RF positivity and need of corticosteroid use due to articular involvement. 

Abstract might be beneficial to include a sentence that briefly summarizes the topic  of the study and to underline the novelty of the study.

It has been added

  •  

The aim of the study was to characterize retrospectively the clinical phenotype of pSS patients with nonidentified ANA profile, in comparison with that of pSS patients with negative ANA, positive typical ANA (anti-Ro/SSA and/or La/SSB) and positive atypical ANA.

Please, improve the description of study aim and the underline the novelty of these observations.

Ok it has been underlined in the intro

  • Data analysis were performed using SPSS Statistics version 13 (IBM, Chicago, Illinois, USA) and GraphPad Prism 8 for Windows (GraphPad Software, San Diego, California, USA). Standard descriptive statistics were used, including percentages, proportions, median and range. Fisher’s exact probability test and Chi-square test were used for contingency tables (qualitative date) and Kruskal–Wallis test was used for continuous data. A p-value < 0.05 was considered statistically significant. To explore the simultaneous relationships between variables, multiple correspondence analysis (MCA) was applied. This technique allows the exploration of the relationships between sub-groups of interest together with the other exploratory variables by comparison of distances and clustering in a multiple dimension space. MCA analysis was generated using SAS software (2020, SAS Institute Inc., Cary, NC, USA).

Please, add some information on statistical tests used to evaluate the data and regarding data presentation.

Additional informations on statistical analysis were added in Material, text and figures.

  •  

Please, improve the description of the figure.

Ok it was improved.

  • Discussion

For the first time to our knowledge, our study has characterized the clinical phenotype of pSS patients with non-identified ANA at diagnosis. In previous studies, ANA positivity was generally used as biomarker regardless of its identification, or the non-identified ANA positive subgroup of pSS patients was excluded (10, 11). 

The discussion section needs to be improved.  It is necessary improve the comparison with published literature.

It was further improved.

  • The limitations of our study lie with the retrospective nature of the study based on the review of medical files and the exhaustiveness of which was left to the discretion of the attending physician; the retrospective calculation of ESSDAI based on the available data, which could lead to its underestimation; and the small numbers of patients in certain subgroups.

Please ameliorate this paragraph I also suggest including the number of patients evaluated for the final analysis as a limit

The paragraph has been ameliorated.

Round 2

Reviewer 2 Report

Comments and Suggestions for Authors

 The manuscript has been improved. I have no further comments.

Comments on the Quality of English Language

Minor changes of English language are required